# Characterizing the Role of SMYD2 in Mammalian Embryogenesis—Future Directions

**DOI:** 10.3390/vetsci7020063

**Published:** 2020-05-12

**Authors:** Dillon K. Jarrell, Kelly N. Hassell, Debbie C. Crans, Shari Lanning, Mark A. Brown

**Affiliations:** 1Department of Bioengineering, University of Colorado Anschutz Medical Campus, Aurora, CO 80523, USA; dillon.jarrell@cuanschutz.edu; 2Cell and Molecular Biology Program, Colorado State University, Fort Collins, CO 80523, USA; kelly.hassell@colostate.edu (K.N.H.); Debbie.Crans@colostate.edu (D.C.C.); 3Department of Chemistry, Colorado State University, Fort Collins, CO 80523, USA; 4Department of Clinical Sciences, Colorado State University, Fort Collins, CO 80523, USA; Shari.Lanning@colostate.edu; 5Epidemiology Section, Colorado School of Public Health, Colorado State University, Fort Collins, CO 80523, USA

**Keywords:** epigenetics, cancer, hematopoiesis, leukemia, histone methylation, vomeronasal organ, gonadotropin-releasing hormone, neural migration, WNT, mesendoderm

## Abstract

The SET and MYND domain-containing (SMYD) family of lysine methyltransferases are essential in several mammalian developmental pathways. Although predominantly expressed in the heart, the role of SMYD2 in heart development has yet to be fully elucidated and has even been shown to be dispensable in a murine Nkx2-5-associated conditional knockout. Additionally, SMYD2 was recently shown to be necessary not only for lymphocyte development but also for the viability of hematopoietic leukemias. Based on the broad expression pattern of SMYD2 in mammalian tissues, it is likely that it plays pivotal roles in a host of additional normal and pathological processes. In this brief review, we consider what is currently known about the normal and pathogenic functions of SMYD2 and propose specific future directions for characterizing its role in embryogenesis.

## 1. Introduction

The 5-member SET and MYND domain-containing (SMYD) family of lysine methyltransferases are broadly expressed throughout the embryo and are essential for a wide range of developmental pathways [1,2,3,4]. These enzymes alter the epigenetic landscapes of cells by catalyzing the transfer of methyl groups to histone lysine residues, thereby modifying chromatin architecture and controlling gene expression during development. In addition, SMYD proteins catalyze methylation of non-histone proteins, thus altering signaling cascades and gene transcription. These multi-domain enzymes catalyze methylation by binding specific enzyme targets in a pocket formed predominantly by the i-SET and core-SET domains and bringing the target into close proximity with the methyl donor s-adenosylmethionine (SAM) in an adjacent pocket (Figure 1c). The second member of the SMYD family, SMYD2, was initially characterized in mice cardiomyocytes during embryogenesis [5]. However, the bulk of research involving SMYD2 has centered on its role in carcinogenesis because of its detected overexpression in a wide range of human cancers (Table 1). Due to its identification as an oncogene, studies investigating SMYD2 in embryogenesis are lacking. We previously detected clear SMYD2 expression in the hypothalamus, liver, kidneys, thymus, vomeronasal organ, and ovaries during development in mice (Figure 1a,b) [5], but the roles of SMYD2 in these organ systems have yet to be fully elucidated and merit further study. In this review, we overview what is currently known about SMYD2 in cancer and embryogenesis and hypothesize new areas for studying SMYD2 in embryogenesis based on preliminary data and analysis of what is currently known. Specifically, we: (1) hypothesize that SMYD2 is essential for the development of mesendoderm and several of its downstream organs (heart, kidney, liver, thymus, and ovaries) in part via its role in WNT signaling; (2) highlight the recently discovered role of SMYD2 in hematopoiesis and leukemias; and (3) suggest a possible role of SMYD2 in the migration of gonadotropin-releasing hormone (GnRH) neurons during the development of the vomeronasal organ (VNO).

## 2. SMYD2 in Cancer

SMYD2 was initially characterized in mouse cardiomyocytes [5,6,7,8], where it was shown to methylate histone H3 at K4 and K36 [9]. However, SMYD2 has received the bulk of its research attention because of its overexpression in several cancers and its interactions with known oncogenes. Overexpression of SMYD2 has been observed in breast, pancreatic, colorectal, esophageal, blood, lung, bladder, and hepatocellular cancers (see refs. in Table 1), and SMYD2-positive tumors are often correlated with poor patient outcomes. These studies have revealed two principal mechanisms by which SMYD2 contributes to carcinogenesis and cancer progression. First, aberrant histone methylation has been associated with many cancer types and results in altered expression levels of several oncogenes. Second, in addition to histone targets, SMYD2 has been shown to directly modulate the activity of cell cycle regulators via post-translational methylation. Studies using human cells both in vitro and in immunodeficient mice demonstrated that SMYD2 represses the activity of tumor suppressers p53 (apoptosis) and RB1 (cell-cycle arrest) via mono-methylation at p53:K370 and RB1:K860 and K810 [8,10,11]. The methylation of AHNAK and AHNAK2—two proteins involved in cell migration and invasion—may also contribute to the role of SMYD2 in carcionogenesis [12]. Other SMYD2 methylation targets include HSP90AB1, ERα, PARP1, PTEN, BMPR2, and β-Catenin [13,14,15,16,17], which is discussed further in Section 3. The known targets of SMYD2, the tissue(s) that are involved, and observed correlations with cancers are summarized in Table 1.

An improved understanding of the specific mechanisms by which SMYD2 contributes to carcinogenesis is likely to reveal new targets for clinical intervention Tremendous efforts by universities and pharmaceutical companies have identified several effective and specific small molecule SMYD2 inhibitors, including AZ505, AZ506, EPZ033294, A-893, BAY-598, LLY-507 [18,19,20,21,22,23,24]. These compounds are promising for the treatment of a wide range of cancers and may also prove to be useful for further investigation of SMYD2 function. While our understanding of the role of SMYD2 in cancers is certainly incomplete and requires further investigation, its function in embryogenesis is even less understood. Further study into the normal roles of SMYD2 in development is necessary for a more-complete understanding of organ development, stem cell differentiation, epigenetics, and SMYD2-mediated pathogenic pathways.

## 3. SMYD2 in Embryogenesis

While our understanding of the role of SMYD2 in cancers remains incomplete, its function in embryogenesis is even less understood. Further study into the normal roles of SMYD2 in development is necessary for a more-complete understanding of organ development, stem cell differentiation, epigenetics, and SMYD2-mediated pathogenic pathways. Towards this understanding, several SMYD2 knockout studies have been conducted both in vitro and in vivo (summarized in Table 2). Of particular note, global SMYD2-knockouts (KO) in zebrafish have revealed severe developmental defects [25,40]; however, mice lacking SMYD2 exhibited no severe developmental phenotypes or life-span defects and in fact gained a reduced susceptibility to leukemias [41]. Our group developed conditional SMYD2 knockouts in mice to study the embryonic role of SMYD2 in heart development [7] and hematopoiesis [35]. These studies are elaborated below in Section 3.1 and Section 3.2, respectively, and serve as a foundation for future investigation of the role of SMYD2 in the development of specific organ systems.

### 3.1. SMYD2 in the WNT Pathway and Mesendodermal Differentiation

SMYD2 expression has been detected in embryonic tissues derived from all three germ layers, including muscle (mesoderm), liver (endoderm), and brain (ectoderm). It is predominantly expressed in the heart during development, however its role in cardiac development remains unclear. Donlin and colleagues [26] demonstrated that SMYD2 forms a complex with the cytoplasmic protein chaperone Hsp90 and the abundant sarcomeric protein titin. Their work revealed that a SMYD2 deficiency resulted in impaired titin stability and altered muscle function. Unexpectedly, however, an Nkx2.5-conditional Smyd2-knockout (cardiac-specific) in mice did not impact heart development [7], suggesting that SMYD2 is not essential for heart development or that its role manifests prior to Nkx2.5 expression in cardiac progenitor cells. Further investigation into the latter conclusion led us to studies revealing a role for SMYD2 prior to Nkx2.5 expression. SMYD2 methylates β-Catenin, which is essential for its nuclear translocation and subsequent activation of WNT signaling [13]. Activation of WNT signaling drives pluripotent stem cell commitment to mesendoderm lineages [40] and is the first step in the differentiation of human pluripotent stem cells (hPSC) to cardiomyocytes in vitro [27,42]. In addition to direct activation of β-Catenin, SMYD2-knockout hPSC lines demonstrated a remarkable reduction of H3K4me1 and H3K36me2 levels at the transcriptional start sites of several signature mesendoderm genes (T, EOMES, MIXL1, and GSC) during differentiation to mesoderm and endoderm [30]. Therefore, we hypothesize that the principal role of SMYD2 in heart development occurs prior to Nkx2.5 expression, which begins 3–4 days after WNT activation via β-Catenin translocation [43]. The role of SMYD2 in the WNT signaling pathway may also explain its presence and importance in other mesendodermal tissues during development, including the liver, kidney, and reproductive systems. Future studies should investigate the specific roles of SMYD2 in the development of these organs using tissue-specific conditional knockout models that allow for ablation of SMYD2 expression at various timepoints during development. Finally, dysregulation of the WNT pathway has been observed in a wide range of cancers, including colorectal, hepatocellular, and breast carcinomas [36,44,45]. The overlap between SMYD2 and the WNT pathway, particularly given their independent associations with such a wide range of cancers, merits immediate further investigation.

### 3.2. SMYD2 in Hematopoiesis and Leukemia

SMYD2 overexpression has been observed in several blood cancers, including B-ALL, T-ALL, CML, MLLr-B-ALL, AML and additional hematopoietic lesions, including CLL and DLBCL [46,47,48,49]. Despite this knowledge, the role of SMYD2 in hematopoeisis was unclear prior to our recent work investigating the effects of SMYD2 deletion in hematopoietic stem cells (HSCs) and their downstream progenitor pools [35]. We discovered that a murine HSC-specific Smyd2 conditional knockout (CKO) yielded a significant decrease in HSC numbers as well as a decrease in some, but not all, downstream myeloid and lymphoid lineages. Deeper investigation revealed that these effects are mediated at least in part by the induction of apoptosis in blood progenitor cells of Smyd2-CKO animals. Additionally, we found that Smyd2-CKO mice exhibited disrupted STAT3 and WNT/β-Catenin signaling in HSC lineages, agreeing with previous studies describing the interactions between SMYD2 and these two proliferation-inducing signaling pathways [13,29].

### 3.3. SMYD2 in the Hypothalamus and Vomeronasal Organ (VNO)

In our initial characterization of SMYD2 in 2006 [5], we sought to determine which embryonic tissues express SMYD2 during development. We performed whole-mount in situ hybridization using murine embryos at day 13.5 with a probe specific to Smyd2 (Figure 1b). While the most pronounced expression was observed in the heart, we also observed clear SMYD2 expression in the hypothalamus and in the vicinity of the vomeronasal organ (VNO). To our knowledge, a possible developmental role of SMYD2 in this region has never been investigated following these initial observations.

Herein, we propose that SMYD2 may play a role in the origin and/or migration of gonadotropin-releasing hormone (GnRH) neurons from the VNO, which are known to emerge from the VNO around E11 and migrate into the basal forebrain between E12–E17 in mice [50,51]. GnRH is released from GnRH neurons located in the hypothalamus in a pulsatile fashion during adolescence, and is essential for proper reproductive function. The release of GnRH represents the initial step in the hypothalamic-pituitary-gonadal signaling axis. Due to the strong SMYD2 signal in the VNO and the hypothalamus during GnRH neural migration between the two developing organs, it is plausible that SMYD2 plays a role in proper neuronal differentiation and/or networking during development. Several factors are implicated in proper GnRH neural migration, including growth factors, extracellular matrix/adhesion molecules, neuro-transmitters, G-protein-coupled receptors, and transcription factors [26,42]. Studies that investigate the roles of histone methylation at transcriptional start sites or direct post-transcriptional methylation in the expression or activity of these factors may elucidate a role for SMYD2 in VNO and hypothalamic development. Besides SMYD2-mediated histone methylation, thorough epigenetic analysis in these regions is likely to further clarify the specific mechanisms that drive VNO development, GnRH neuron migration, and neural networking in the hypothalamus.

## 4. Conclusions and Future Directions

Methyltransferase enzymes have recently been recognized as key players in modulating gene expression and protein activity in a wide range of physiologic processes. The SMYD family of methyltransferase enzymes has been shown to be particularly important during embryogenesis via histone methylation, and have also been shown to regulate the expression and activity of several known oncogenes. We previously demonstrated that the second member of the SMYD family, SMYD2, is widely expressed during normal embryogenesis but also in several cancers. To further elucidate the role of SMYD2 in development and carcinogenesis, we propose four areas for future study.

First, the purpose of the robust expression of SMYD2 in the heart during development has yet to be uncovered. It is known that SMYD2 expression is dispensable in early cardiac progenitor cells expressing Nkx2.5 [7], which may be explained by compensation from SMYD1 or by an earlier role for SMYD2 during heart development. Experimentation seeking to elucidate an earlier role for SMYD2 in heart development would be valuable in answering this question.

Second, SMYD2-WNT-β-catenin signaling merits deeper investigation in two contexts. First, WNT signaling is essential for mesendoderm commitment, which could explain the role of SMYD2 in the development of a wide range of downstream organs (heart, kidneys, thymus, liver, and ovaries) and represents a reasonable explanation for the role of SMYD2 in heart development prior to Nkx2.5 expression. Second, this signaling axis is essential for hematopoetic stem cell (HSC) renewal and is disrupted in a wide range of HSC-derived blood cancers. Targeting the SMYD2-WNT-β-catenin pathway in HSCs may provide a successful approach for future cancer therapies.

Third, while our recent work established SMYD2 as an important mediator in leukemias [35], the specific mechanisms by which it directs apoptosis in HSCs and downstream progenitor pools remain unclear. Further study into the roles of SMYD2 in normal and pathogenic blood cell renewal and differentiation is likely to reveal important transcriptional leukemic targets for clinical intervention.

Fourth, SMYD2 is clearly expressed near the vomeronasal organ (VNO) during development but its role has yet to be studied in this organ system [5], Figure 1b. Due to the parallel expression in the hypothalamus during GnRH neuron migration from the VNO, we postulate that SMYD2 may play a key role in the proper migration and networking of these neurons from the VNO to the hypothalamus.

The broad expression patterns of SMYD2 in embryogenesis and carcinogenesis have yet to be completely explained. Further investigation of the functions of SMYD2 in these contexts will expand our understanding of mammalian development and improve our clinical management of cancer.

## Figures and Tables

**Figure 1 vetsci-07-00063-f001:**
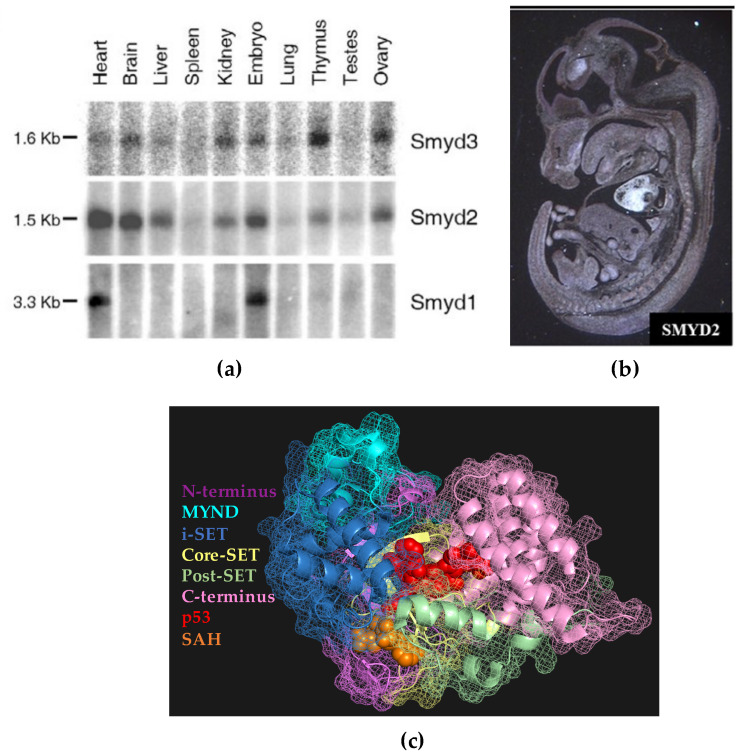
Expression patterns of SMYD2. (**a**) Northern blot displaying tissue-specific mRNA expression levels of SMYD1, SMYD2, and SMYD3 during mouse development. Middle panel: SMYD2 is widely expressed in the heart, brain, liver, kidney, thymus, and ovary; (**b**) Murine E13.5 sagittal section with an anti-sense SMYD2 label depicting broad SMYD2 expression in the developing embryo, especially in the heart, hypothalamus, and vomeronasal organ; (**c**) the color-coded domains of SMYD2, common to all five SMYD proteins. Panel (**a**) used with permission from Ref. [5] according to Creative Commons Attribution License (http://creativecommons.org/licenses/by/2.0). Panel (**c**) created using PyMol software and the protein data bank file 3TG5 [6].

**Table 1 vetsci-07-00063-t001:** SMYD2 target proteins, functions, and associated cancers.

Target Protein	Protein Function	Associated Cancers	Refs.
Heat shock protein 90 (HSP90)	Titin stability and sarcomere assembly in muscle tissue, drives handling of several oncoproteins	Lung, breast, urothelial, colorectal, bladder carcinomas	[25,26,27,28]
Signal transducer and activator of transcription-3 (STAT3)	Becomes activated in complex with SMYD2, p65, and NF-kB, transcription activator, feed-forward SMYD2 expression, increased proliferation, stem cell self-renewal	Breast carcinomas, leukemias	[29]
Histone 3 (K4)	Interacts with RNA Polymerase II, RNA helicase, drives gene transcription at promotor regions	Lung, breast, glioblastomas	[7,28,30]
Histone 3 (K36)	Associated with gene bodies, defines exons, recruits HDACs, modulates transcription	Embryonic kidney and fibroblast carcinomas	[5,28,31,32,33]
p53 (K370)	Cell cycle arrest	p53 dysfunction detected in most cancer types	[7,8,31,33,34]
β-catenin (K133)	Methylation promotes nuclear translocation and activation of WNT signaling	B-cell precursor acute lymphoblastic leukemia (ALL), colon, breast, and hepatocellular carcinomas	[13,35,36]
Retinoblastoma tumor suppressor protein (RB1; K810)	Promotes cell cycle progression; G1/S transition	Bladder carcinomas	[10,11]
Ahnak nucleoprotein 2	Cell migration and invasion	Chronic myeloid leukemia (CML), Mixed lineage leukemia rearranged adult B-acute lymphoblastic leukemia (MLLr-B-ALL), adult B-acute lymphoblastic leukemia (B-ALL), and T-cell acute lymphoblastic leukemia (T-ALL)	[18,19,37]
Estrogen receptor-α (ER-α; K266)	Nuclear receptor for sex hormone estrogen, potent upstream activator of gene transcription	Breast	[17]
Poly-(ADP-ribose) polymerase 1 (PARP1; K528)	Enhances poly (ADP-ribose) activity	Esophageal squamous cell and bladder carcinoma, pediatric acute lymphoblastic leukemia	[16,31,33]
Phosphatase and tensin homologue (PTEN; K313)	Down-regulates tumor suppressor & activates the phosphatidylinositol 3-kinase-AKT pathway	Prostate, small cell lung cancers	[15]
Bone Morphogenic Protein Receptor-2 (BMPR2)	Methylation upregulates BMP2 signaling, important for bone development and MSC proliferation	Skin, colorectal, ovarian, breast, embryonic kidney carcinomas	[13]
Frizzled-2	STAT3 interaction, epithelial-mesenchymal transition, WNT and Hippo signaling	Hepatocellular	[35,38]
Erythrocyte membrane protein band 4.1 like (3EBP41L3; K1610)	Tumor suppressor	Breast, non-small-cell lung, brain, ovarian cancers	[39]

**Table 2 vetsci-07-00063-t002:** SMYD2 knockout studies and observed developmental phenotypes.

Model	Phenotype	Refs.
*Nkx2.5 Cre/Smyd2^flox/flox^* (CKO) mice (cardiac-specific)	No phenotype	[7]
*Mx-1Cre/Smyd2^flox/flox^* (CKO) mice (hematopoetic stem cell-specific)	Reduced HSC and downstream progenitor populations via apoptosis and cell cycle blocking	[35]
*Smyd2^-/-^* human embryonic stem cells	Blocked mesendodermal lineage commitment, No affect on self-renewal and neurectoderm	[30]
*morpholino-Smyd2a^-/-^* zebrafish	Developmental delay, tail abnormalities, over-induction of Nodal pathway	[40]
*morpholino-Smyd2a^-/-^* zebrafish	Severe cardiac structural and sarcomeric malformation, tail abnormalities	[25]
*Smyd2^-/-^* (global KO) mice	No developmental phenotype, reduced susceptibility to leukemia	[41]
CRISPR/Cas9 *SMYD2* KO human keratinocytes (*in vitro*)	Impaired BMP2 morphogen signaling	[14]

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
