# Peer review of "Characterizing the Role of SMYD2 in Mammalian Embryogenesis—Future Directions"

_vetsci, 2020, doi:10.3390/vetsci7020063_

Round 1

Reviewer 1 Report

The manuscript entitled “Characterizing the role of SMYD2 in embryogenesis – Future direction” reviews the functions of the lysine methyltransferase SMYD2 in embryogenesis and tumorigenesis. The review is clear, well presented and written. It focuses on the role of SMYD2 in mesoderm development and in cancer.

The review is extremely brief and perhaps sometime too succinct. For instance, the authors states (page 2) “Small molecule SMYD2 inhibitors (?) have already been shown to be effective in vitro and are promising for the treatment of a wide range of cancer.” Only one sentence and one reference [17] to sum up the discovery of a number of compounds including AZ505, A-893, BAY-598, LY-507 and a lot of efforts made in particular by several big companies from the pharmaceutical industry.

I believe the author should develop a bit more the different points of the review and should go into details that are more mechanistic concerning the role of SMYD2 in cancer.

Author Response

Dear Reviewer,

Our sincerest thanks for your time in reviewing our manuscript. We have carefully considered your two main points of revision: the brevity of some sections and expanding upon the mechanistic role of SMYD2 in cancer.

We have indeed expanded the section regarding small molecule SMYD2 inhibitors, and have added more references. We also looked for other areas in the text that were too brief, and decided to also expand the section describing what is currently known about SMYD2 in embryogenesis. We understand that the review remains brief, but this is intentional. Our main purpose in writing this review is to quickly suggest 3-4 specific areas for future investigation of SMYD2 in embryogenesis.

With regard to expanding the mechanistic descriptions of SMYD2 in cancer, we have made no changes. The primary purpose of this review is to suggest areas for future study of SMYD2 in embryogenesis, not in cancer. We included the section on SMYD2 in cancer (and Table 1) only to briefly inform the readers of the importance of SMYD2 in cancer and to introduce the broad expression of SMYD2 targets throughout the body. We believe that the two mechanisms described in Section 2 are sufficient for the context of this review.

We hope that you find these revisions appropriate. Thank you again for your time and diligence in your review.

Reviewer 2 Report

The review is comprehensive, well organized and include main references and available information. Congratulation to authors. I miss some more specific references to the species in the different studies reported. The lack of line and page numbers difficult the review of the manuscript.

Some minor points:

Tittle: “mammalian embryogenesis”

Expanded report:

The authors review the role of SMYD2, a SET and MYND Domain-containing (SMYD) family of lysine methyltransferases, on mammalian embryogenesis. Despite SMYD2 is mainly expresses in cardiac tissue its role in heart development has not been elucidated. The ubiquitous expression of SMYD2 in a wide range of tissues has led authors to review current knowledge about the role of SMYD2 in embryogenesis.

 The review is comprehensive, well organized and include main references and available information. However, for a better understanding it would be desirable to mention the species where the different studies reported have been performed.

Minor points:

Tittle: specify “mammalian embryogenesis”

Ln 35-37: which species are authors referring to?

Ln 55: mice cardiomyocytes?

Ln 4-75: I agree. That is why a specific section on “SMYD2 function in embryogenesis” or a table summarizing the knowledge and specific studies done with embryo would be a plus for the readers.

Lns. 73 and 94: in vitro (consistency on cursive)

Table 1: for consistency expand all acronyms. Signal transducer and activator of transcription 3 (STAT3), AHNAK2, PARP1, …

Author Response

Dear Reviewer,

Our sincerest thanks for your revisions and comments.

We have added “mammalian” to the title.

Throughout the manuscript, we have added in the experimental models used (animal type, cell type, etc.). Thank you for this suggestion.

We have indeed added a table (Table 2) and paragraph describing the current state of knowledge surrounding SMYD2 in embryogenesis. We agree that these sections make the paper more coherent and complete.

We have corrected the inconsistent italics.

We have expanded all of our acronyms in Table 1.

We hope that you find these revisions appropriate. Thank you again for your time and diligence in your review.

Reviewer 3 Report

The manuscript entitled “Characterizing the role of SMYD2 in Embryogenesis-Future Directions” deals with topics of function of SMYD2, a histone methyltransferase, in cancers and tissue differentiation process. Authors had good coverage of current topics of SMYD2 studies in this review manuscript.

In Figure 1, authors display expression profile of SMYD1, SMYD2 and SMYD3. In my opinion, it is beneficial for readers that if authors add some graphic scheme of domain structure of SMYD proteins.

Although authors display and state broad expression of SMYD2 in mouse tissues, they only deal with specific topics such as mesendoderm (section 3), hematopoiesis (section 4) and hypothalamus (section 5). I recommend authors deal with studies of (conditional) SMYD2 knockout mice and its phenotype.

In section 5, authors propose that SMYD2 plays a role in the origin and/or migration of GnRH neurons from VNO. To state that, it is required to reveal not only expression of SMYD2 in VNO but also show or refer evidence such that there are few migration of GnRH neurons in SMYD2 knockout mice.

Author Response

Dear Reviewer,

Our sincerest thanks for your time in reviewing our manuscript. We have carefully considered your three primary points of revision: a figure/description of SMYD domain structure, a section considering cKO SMYD2 knockout mice, and the bold claims involving the VNO.

We have indeed added a panel to figure 1 depicting the structural domains of SMYD2. Additionally, we have mentioned these domains in the Introduction.

We have added Table 2 and a short paragraph summarizing SMYD2 cKO studies. We already mentioned several of these studies in the text, but agree that it is helpful to make very clear what is already known about SMYD2 in development.

With regard to our claims regarding SMYD2 in the VNO, we revised our wording throughout the manuscript to emphasize the fact that we are merely proposing a possible role because of observed SMYD2 expression in the vicinity of the VNO at the time of GnRH neuron migration. We are not claiming to have any other evidence to support this, and are not claiming that it is true. We are simply suggesting that it should be tested. The evidences that you mentioned would certainly be part of such a study.

We hope that you find these revisions appropriate. Thank you again for your time and diligence in your review.

Reviewer 4 Report

 Jarrel et al. review about SMYD2 contains all relevant information. However, the title of this review is misleading. Also, the chapters are not logically organized and thus the flow of information is disturbed. Authors should reorganize the chapters or make links between the mentioned information.

Author Response

Dear Reviewer,

Our sincerest thanks for your time in reviewing our manuscript. We have carefully considered your two primary points of revision: the title and the overall organization.

While we certainly spend a significant portion of the manuscript reviewing SMYD2 in cancer rather than embryogenesis, we believe that the primary novelty of the review is captured well in the current title. To add a greater emphasis on embryogenesis in the text (and thus make the title more appropriate), we have added an additional table overviewing the currently-known roles of SMYD2 specifically in embryogenesis.

With regard to the paper’s organization, we have re-numbered the sections to make it clear that the sections involving the WNT pathway, Hematopoeisis, and VNO development all fall under the category of “SMYD2 in Embryogenesis”. In addition, we have added a short introduction paragraph to this section so that it is very clear that we are turning the reader’s attention away from SMYD2 in cancer and towards SMYD2 in development.

We hope that you find these revisions appropriate. Thank you again for your time and diligence in your review.

Round 2

Reviewer 1 Report

The authors answered to my concerns in the revised manuscript. I have no further comments.